# RELAXED ATTENTION FOR TRANSFORMER MODELS

## ABSTRACT

The powerful modeling capabilities of all-attention-based transformer architectures often cause overfitting and—for natural language processing tasks—lead to an implicitly learned internal language model in the autoregressive transformer decoder complicating the integration of external language models. In this paper, we explore relaxed attention, a simple and easy-to-implement smoothing of the attention weights, yielding a two-fold improvement to the general transformer architecture: First, relaxed attention provides regularization when applied to the self-attention layers in the encoder. Second, we show that it naturally supports the integration of an external language model as it suppresses the implicitly learned internal language model by relaxing the cross attention in the decoder. We demonstrate the benefit of relaxed attention across several tasks with clear improvement in combination with recent benchmark approaches. Specifically, we exceed the former state-of-the-art performance of 26.90% word error rate on the largest public lip-reading LRS3 benchmark with a word error rate of **26.31%**, as well as we achieve a top-performing BLEU score of **37.67** on the IWSLT14 (DE → EN) machine translation task without external language models and virtually no additional model parameters. Code and models will be made publicly available.

## 1 INTRODUCTION

Early encoder-decoder models emerged from machine translation, where the encoder compressed the entire source language sentence into a fixed-length embedding vector (Cho et al., 2014b). This is particularly difficult for very long sentences (Cho et al., 2014a), as the fixed-length embedding vector is only a limited-capacity representation. The use of attention, introduced in Bahdanau et al. (2015), enabled the computation of variable-length weight distributions over the input sequence and soon turned out to be advantageous for far more applications than just neural machine translation (NMT), e.g., automatic speech recognition (ASR) (Chorowski et al., 2015; Chan et al., 2016; Bahdanau et al., 2016), language modeling and understanding (Devlin et al., 2019), object detection (Carion et al., 2020), and image classification (Dosovitskiy et al., 2021; Liu et al., 2021b). Soon the most prominent attention-based encoder-decoder (AED) model emerged, namely the transformer (Vaswani et al., 2017) model. Without the use of any recurrency, it entirely relies on self-attention in the encoder to model temporal dependencies in the input and cross attention in the decoder to extract relevant timesteps thereof during the autoregressive decoding process. While transformers in language modeling tasks are well-suited for upscaling the model size and depth without any saturation when large amounts of data are present (Devlin et al., 2019; Brown et al., 2020; Kaplan et al., 2020; Fedus et al., 2022), they are also susceptible to overfit and require strong regularization to learn at all (Xu et al., 2021; Popel & Bojar, 2018; Steiner et al., 2021). In a study exclusively on ASR (Lohrenz et al., 2021), it was shown that regularization by smoothing the attention weights in the decoder's cross attention, dubbed relaxed attention, improves performance when the transformer model is combined with an external language model but, for reasons yet to be explored, does not help without a language model.

In this work, we take on the idea of relaxed attention to expand it to the self-attention layers in the encoder, regularizing already the encoder. Thereby, we increase the method's versatility as it becomes applicable to encoder-only transformers, which are common in several non-sequence tasks such as image classification or pre-trained bidirectional encoder representation by transformer (BERT, Devlin et al. (2019)) models. Our main contributions are summarized as follows:

- We introduce relaxed self-attention in the transformer *encoder* to improve generalization and develop fuzzy relaxation as a variant thereof.
- Beyond relaxed self-attention, we extensively investigate the capability of relaxed cross attention in the *decoder* of sequence-to-sequence transformer models and show that the improvement is due to better external language model integration as it suppresses the influence of the internal language model.
- We show improvements of the relaxed attention approaches on a variety of tasks including automatic speech recognition, lip-reading, machine translation, and image classification. On the lip-reading and machine translation task we report a new state of the art and top-performing result, respectively.

The paper is structured as follows: After a summary of related work in Section 2, we introduce the relaxed attention approach in Section 3, followed by the experimental evaluation including results and discussion in Section 4. Section 5 concludes the paper.

## 2 RELATED WORK

**Regularization of transformer models**   In this work, we introduce a regularization method to the self-attention function (Vaswani et al., 2017), which is fundamental to transformer models. Several regularization approaches proposed for such networks in the past are related to the network output of transformer models by modifying the loss computation, either through label smoothing (Müller et al., 2020), or by introducing additional loss terms. This could be a CTC-loss computed on the encoder outputs (Karita et al., 2019; Chen et al., 2021) for monotonous tasks such as ASR, or a divergence term between output softmax distributions of two forward passes with different dropout masks (Liang et al., 2021; Shen et al., 2020). Related to the network input, several—mostly application-dependent—data augmentation approaches such as spectral augmentation for ASR (Park et al., 2019b), or cutoff for machine translation (Shen et al., 2020) have proven to be effective. Another set of regularization methods, specific to transformer models, adds a loss term to encourage attention heads to yield diverse outputs (Li et al., 2018; Audhkhasi et al., 2022) or is based on the dropout technique (Srivastava et al., 2014) and randomly masks attention heads (Zhou et al., 2020; Sun et al., 2020) or entire en-/decoder block layers (LayerDrop) (Fan et al., 2020) during training. It was also observed that only a subset of specialized attention heads contribute to model performance, while other heads remain useless and can be pruned (Voita et al., 2019). Relaxed attention in Lohrenz et al. (2021) was used to prevent too narrow attention weight distributions in the cross attention during training which only yielded improvements with an external language model in ASR. We, however, apply this approach to the self-attention function to reduce over-fitting already in the encoder and investigate if relaxed self-attention also helps when applied during both training and test (matched inference). In addition we include a variety of the aforementioned—proven to be effective—regularization methods as baselines and show that relaxed attention is able to further improve performance yielding complementarity to other regularization methods.
When attention-based encoder-decoder networks were first applied to ASR, Chorowski et al. (2015) proposed a modified softmax function to smooth the attention weights in the cross attention between encoder and decoder by replacing the exponential function in the standard softmax function with a sigmoid. Thereby, they compressed the probability-like outputs, but didn't take into account the input sequence length, despite the authors' observation that longer sentences require less smoothing of the attention weights. Even though this method dubbed *smooth focus* was so far only applied to recurrent neural network (RNN)-based AED models, we include it as a reference method in our simulations as it is the closest to the relaxed attention approach.

**Internal language model handling**   For many sequence-to-sequence tasks the integration of language models (LMs) to AED models is of dual use: First, LMs leverage the use of large amounts of additional text-only data to improve performance. Second, LMs can be utilized to adapt acoustic models to domains which differ from the original acoustic training data domain. Several techniques exist to combine language models with AED models, such as shallow fusion (Gülçehre et al., 2015), deep fusion (Gülçehre et al., 2015), and cold fusion (Sriram et al., 2018), whereas shallow fusion still is the most common solution due to its simplicity and flexibility. However, AED models tend to learn an internal language model in the autoregressive decoder (McDermott et al., 2019), which can either be suppressed by subtracting an additional LM trained only on text transcriptions from the

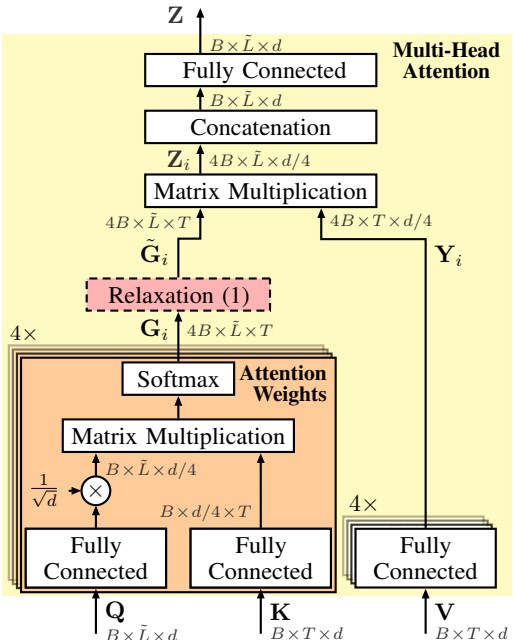

Figure 1: Multi-head attention (MHA) as used in encoder and decoder blocks of transformer models with $N_{\mathrm{h}} = 4$ attention heads. The proposed **relaxed attention** (red block) is presented in Section 3.

acoustic training data (e.g., density ratio fusion from McDermott et al. (2019)) or—as was shown more recently—can be adapted to a new domain (Meng et al., 2022) requiring additional retraining. For the specific application of automatic speech recognition, in a small study, Lohrenz et al. (2021) have investigated relaxed cross attention, whereby performance improvements were only achieved with external language models. In our work, we investigate the hypothesis that relaxed cross attention successfully suppresses the internal language model, which in contrast to the aforementioned methods does not—apart from a single hyperparameter—require any additional models (McDermott et al., 2019), parameters (Sriram et al., 2018), or adaptation trainings (Meng et al., 2022), but weakens the internal language model during training of the transformer thus supporting shallow fusion (Gülçehre et al., 2015). In addition, we will introduce relaxed self-attention, which improves performance in many applications even without the use of an explicit LM.

## 3 PROPOSED RELAXED ATTENTION

Scaled dot-product multi-head attention (MHA, see Figure 1) is typically used in two variants in the encoder-decoder transformer model (Vaswani et al., 2017): First, it is used in the *en*coder as self-attention to model positional (e.g., temporal) dependencies in the preprocessed input sequence indexed with $t \in \{1, \ldots, T\}$ of length $T$. Second, it is used in the *de*coder as cross attention (also often referred to as encoder-decoder or source target attention), which draws the decoder's attention to relevant parts in the encoded input sequence $\mathbf{h}_1^T \in \mathbb{R}^{T \times d}$ for decoding at output sequence index $\ell \in \{1, \ldots, L\}$ with model dimension $d$. In case of self-attention[1], all MHA inputs (key $\mathbf{K}$, value $\mathbf{V}$, query $\mathbf{Q}$) are the same, i.e., $\mathbf{K} = \mathbf{V} = \mathbf{Q}$, with query input $\mathbf{Q} \in \mathbb{R}^{\tilde{L} \times d}$ of length $\tilde{L} = T$. For cross attention, key and value inputs, $\mathbf{K} \in \mathbb{R}^{T \times d}$ and $\mathbf{V} \in \mathbb{R}^{T \times d}$, respectively, stem from the encoder output $\mathbf{h}_1^T$ yielding $\mathbf{K} = \mathbf{V} = \mathbf{h}_1^T$, while the query input $\mathbf{Q}$ comes from the previous decoder layer with $\tilde{L} = 1$ during inference and $\tilde{L} = L$ during training, where for the latter all $L$ tokens of the target sequence are processed in parallel. Details of the entire typical encoder-decoder transformer architectures are recapitulated in Appendix A.1. The attention weights $\mathbf{G}_i(\mathbf{Q}, \mathbf{K}) \in \mathbb{I}^{\tilde{L} \times T}$ for the scaled dot-product MHA sum up to one across the query input length $\tilde{L}$ after the softmax activation function and thus

---

[1]Note that *masked* self-attention is also used in the *de*coder to attend to prefix output tokens (cf. Fig. 2, Appendix A.1), while in this work we focus on the multi-head attention variants that attend to the input domain.

can be interpreted as a probabilistic weighting applied to the value input projection $\mathbf{Y}_i \in \mathbb{R}^{T \times d/N_{\mathrm{h}}}$, with $N_{\mathrm{h}}$ being the number of attention heads each indexed with $i \in \mathcal{N}_{\mathrm{h}}$.

Relaxed attention follows the basic principle of regularization by introducing some stress into the training process. Shown as red box in Figure 1 it modifies the standard attention weights $\mathbf{G}_i(\mathbf{Q}, \mathbf{K})$ that draw attention to the encoded input sequence. Our *relaxed* attention weights for scaled dot-product attention are defined as simple as

$$\tilde{\mathbf{G}}_i(\mathbf{Q}, \mathbf{K}) = \left[ (1 - \gamma)\mathbf{G}_i(\mathbf{Q}, \mathbf{K}) + \gamma \frac{\mathbf{1}}{T} \right], \;\; i \in \mathcal{N}_{\mathrm{h}}, \tag{1}$$

gradually injecting a uniform distribution (with $\mathbf{1}$ here being an $\tilde{L} \times T$ matrix of ones), thereby smoothing the distribution of attention weights and imposing some of the attention weight mass across the entire input sequence. Please note that the normalizing division by $T$ in equation (1) is different for each file and minibatch for sequence tasks due to varying input sequence lengths. This provides a natural and rich variation of the effective height of the injected uniform distribution and prevents compensation of the relaxation by the learning process. The injection of the uniform distribution is controlled by a relaxation coefficient $\gamma \in [0, 1]$, which is a constant single hyperparameter for all respective attention layers.

While Lohrenz et al. (2021) only investigated relaxed *cross* attention, only for automatic speech recognition and only during training, in our work (i) we propose relaxed cross attention and self-attention, (ii) during training and during inference (*matched inference*), (iii) we investigate their application to automatic speech recognition, lip-reading, machine translation, and image classification, and (iv) we introduce *fuzzy relaxation* for the image classification task, where we randomly draw the relaxation coefficient from a normal distribution $\gamma \sim \mathcal{N}(x; \mu = \gamma_0, \sigma^2)$, with the initially set $\gamma_0$ being the mean $\mu$. For this specific task, the *variable* sequence length $T$ in equation (1) is substituted by a *constant* number of image patch tokens $M^2$ (see equation (3) in Appendix A.2), thereby omitting the aforementioned natural variation. Fuzzy relaxation re-establishes this variation of the relaxation by randomizing $\gamma$ during training, while for the matched inference case, the relaxation coefficient is kept fixed at $\gamma = \mu = \gamma_0$ during inference. Details for the encoder-only transformer used for the image classification task are given in Appendix A.2.

## 4 EXPERIMENTAL VALIDATION AND DISCUSSION

### 4.1 APPLICATION TO AUTOMATIC SPEECH RECOGNITION

**Task and datasets** Automatic speech recognition transforms recorded speech signals into a sequence of text tokens. We investigate our relaxed attention method on the Librispeech dataset (Panayotov et al., 2015) with the `clean` and `other` conditions of the `dev` and `test` subsets. We measure system performance in terms of word error rate $\mathrm{WER} = 1 - \frac{N - D - I - S}{N}$, depending on the number of words $N$, deletions $D$, insertions $I$ and substitutions $S$. All raw speech signals are sampled at 16 kHz and analyzed with a 25 ms window at a frame shift of 10 ms. As common in ASR, we also use an external language model trained on the text labels of the 960 h training set as well as on the text-only Librispeech language model training corpus, the latter containing sentences from a total amount of 14,500 books from project Gutenberg (Panayotov et al., 2015) which are accessible under public-domain. The Librispeech ASR corpus is available under the very permissive Creative Commons BY 4.0 license.

**Models and training** For training with 100 h and 960 h of training data, we trained standard encoder-decoder transformer models (Vaswani et al., 2017) from scratch in the small and `base` configuration, comprising 19.3M and 69.8M parameters, respectively. As common for ASR, filterbank features are extracted for each time frame $t$ and then preprocessed by a four-layer convolutional neural network, each using $3 \times 3$ filter kernels (cf. preprocessing block in Figure 2, Appendix A.1). All hyperparameters were set according to the recipes available in the `fairseq` based `espresso` toolkit[2] (Wang et al., 2019) except the relaxation coefficients $\gamma$, which have been tuned on the joint `clean` and `other` portions of the `dev` set for both, relaxed cross attention and relaxed self-attention. As additional regularization we use SpecAugment (Park et al., 2019a), label smoothing (Müller et al., 2020) and dropout (Srivastava et al., 2014) during training. See Appendix A.3.1 for more details.

---

[2]ASR training recipes at **https://github.com/freewym/espresso**

| Training data | Approach | *without* LM | | | | *with* LM | | | |
|---|---|---|---|---|---|---|---|---|---|
| | | dev | | test | | dev | | test | |
| | | clean | other | clean | other | clean | other | clean | other |
| 100 h | Baseline (Lohrenz et al. (2021), resim.) | 13.98 | 28.71 | 14.82 | 29.31 | 10.62 | 24.19 | 12.06 | 25.56 |
| | + smooth focus (Chorowski et al., 2015) | 14.60 | 28.73 | 15.50 | 30.78 | 10.83 | 24.86 | 12.11 | 26.46 |
| | + relaxed cross attention | 13.91 | 28.70 | 14.70 | 30.10 | **9.33** | **22.16** | **10.62** | **23.04** |
| | + matched inference | 14.30 | 29.03 | 15.15 | 30.09 | 11.04 | 25.19 | 12.16 | 26.36 |
| | + relaxed self-attention | 13.48 | **27.87** | **14.20** | **28.96** | 10.22 | 23.53 | 11.04 | 24.55 |
| | + matched inference | **13.43** | 28.00 | 14.46 | 29.23 | 10.01 | 23.96 | 11.19 | 25.32 |
| 960 h | Baseline (Lohrenz et al. (2021), resim.) | 3.92 | 9.00 | 4.47 | 9.23 | 3.73 | 8.52 | 4.40 | 8.95 |
| | + smooth focus (Chorowski et al., 2015) | 4.11 | 9.42 | 4.35 | 9.63 | 3.70 | 9.18 | 4.31 | 9.33 |
| | + relaxed cross attention | 3.95 | 9.33 | 4.28 | 9.45 | **3.44** | **7.74** | **3.58** | **8.35** |
| | + matched inference | 3.96 | 9.29 | 4.20 | 9.40 | 3.69 | 8.95 | 4.21 | 9.46 |
| | + relaxed self-attention | 3.82 | **8.50** | **4.05** | **8.71** | 3.52 | 8.03 | 4.17 | 8.51 |
| | + matched inference | **3.79** | 9.12 | 4.09 | 9.07 | 3.35 | 8.28 | 3.91 | 8.50 |

Table 1: **Automatic speech recognition** results in terms of WER (%) on the **Librispeech** task using standard **encoder-decoder transformer** models. Attention relaxation is applied in training only, except for "matched inference" (attention relaxation in training *and* test). We separately use the 100 h and 960 h training datasets and highlight the respective best results for each size in **bold** font.

**Results and discussion**    For both, the 100 h and 960 h training data cases in Table 1, the resimulated baselines (using training scripts from Wang et al. (2019)) yield similar results as in Lohrenz et al. (2021) using a standard transformer approach. The smooth focus method (Chorowski et al., 2015) has a higher WER compared to the baseline on the small training data case, but yields small improvements on some `clean` settings for the 960 h training data case. Compared to smooth focus, relaxed self- *and* cross attention adapt to the length $T$ of the input sequence, with the latter yielding solid WER reduction across all `dev` and `test` conditions when an LM is used (right-hand side of Table 1), thereby confirming the results of Lohrenz et al. (2021). In Appendix A.4, we show that the strong improvement with LM using relaxed cross attention is due to improved internal language model suppression. Without an LM, both the resimulated baseline *and* relaxed cross attention approaches are outperformed by our new relaxed self-attention in all `dev` and `test` conditions for both training data cases. Specifically, the WER across the `test` conditions of the 960 h case for relaxed self-attention improved by a relative 9% (`clean`) and 5% (`other`) compared to the resimulated baseline, yielding complementary regularization of our method to the other employed regularization methods. Note that in all aforementioned cases, *relaxed attention is best when used only in training*. Only in a very specific case on the `dev` set, however, "matched inference", i.e., relaxed self-attention in training *and* test, is slightly ahead of using it in training only. Please also see the Appendix, where we conduct deeper analysis on initialization seed robustness (Appendix A.5), sensitivity of the relaxation coefficient $\gamma$ (Appendix A.7), as well as an ablation on the related attention dropout (Appendix A.6).

## 4.2    APPLICATION TO LIP-READING

**Task and datasets**    Automatic lip-reading strives to process an image sequence from recordings of talking faces. We evaluate lip-reading performance in terms of WER on the test partition of the Lip Reading Sentences 3 (LRS3) dataset consisting of a total of 1,321 recorded videos of English utterances sourced from TED talks (Afouras et al., 2018). To investigate the performance of the relaxed attention approach on recently successful self-supervised learning approaches, we closely follow the training setup from Shi et al. (2022) and use audio-visual hidden unit BERT (AV-HuBERT) encoder models pre-trained on the English subset of the Voxceleb2 dataset (Chung et al., 2018), containing a total amount of 1,326 hours of unlabeled video recordings. For some experiments we also use an external language model trained on the joint text data from LRS3 and the Librispeech language model training corpus. LRS3 is publicly available under the TED terms of use as well as the Creative Commons BY-NC-ND 4.0 license.

| Unlabeled data (pre-training) | Labeled data (fine-tuning) | Approach | *without* LM | | *with* LM | |
|---|---|---|---|---|---|---|
| | | | dev | test | dev | test |
| 334 h | 433 h | Afouras et al. (2020) | — | — | — | 59.80 |
| — | 1,362 h +157 h | Afouras et al. (2018) | — | 59.90 | — | 58.90 |
| — | 433 h + 157 h | Xu et al. (2020) | — | 57.80 | — | — |
| — | 433 h | Ma et al. (2021) | — | — | — | 46.90 |
| — | 433 h + 157 h | Ma et al. (2021) | — | — | — | 43.30 |
| — | 33,000 h | Makino et al. (2019) | — | 33.60 | — | — |
| | | Baseline (Shi et al. (2022)) | — | 46.10 | — | — |
| | 30 h | Baseline (Shi et al. (2022), resimulated) | 47.36 | 45.90 | 47.61 | 45.33 |
| | | + smooth focus (Chorowski et al. (2015)) | 47.08 | 45.80 | 46.69 | 45.38 |
| | | + relaxed cross attention | **45.92** | **44.00** | **45.11** | **42.68** |
| | | + matched inference | 46.55 | 45.25 | 46.46 | 45.39 |
| | | + relaxed self-attention | 46.90 | 45.47 | 46.95 | 44.64 |
| | | + matched inference | 46.85 | 45.04 | 46.68 | 44.69 |
| | | Baseline (Shi et al. (2022)) | — | 28.60 | — | — |
| 1,326 h | 433 h | Baseline (Shi et al. (2022), resimulated) | 21.90 | 29.52 | 21.61 | 28.97 |
| | | + smooth focus (Chorowski et al. (2015)) | 21.87 | 29.25 | 21.29 | 28.86 |
| | | + relaxed cross attention | 22.12 | 29.49 | **21.05** | **28.05** |
| | | + matched inference | 22.11 | 29.20 | 21.55 | 28.55 |
| | | + relaxed self-attention | 21.89 | 28.96 | 21.25 | 28.55 |
| | | + matched inference | **21.86** | **28.84** | 21.24 | 28.48 |
| | | Baseline (Shi et al. (2022)) | — | 26.90 | — | — |
| | 433 h + 1,326 h | Baseline (Shi et al. (2022), resim.) | 17.71 | 26.73 | 17.18 | 26.50 |
| | | + smooth focus (Chorowski et al. (2015)) | 17.42 | 26.78 | 17.22 | 26.29 |
| | | + relaxed cross attention | **17.40** | 26.57 | **16.92** | **25.51** |
| | | + matched inference | 17.71 | 26.43 | 17.48 | 25.95 |
| | | + relaxed self-attention | 17.54 | **26.31** | 17.12 | 26.17 |
| | | + matched inference | 17.65 | 26.40 | 17.16 | 26.06 |

Table 2: **Automatic lip-reading** results in terms of WER (%) on the **LRS3** task using various sequence-to-sequence topologies (top segment baselines) or `AV-HuBERT` encoders (lower three segments) pre-trained on unlabeled English data from Voxceleb2 and fine-tuned with a joint transformer decoder on the given amount of fine-tuning training data. We also use self-training (bottom segment) by creating pseudo-labels for the 1,326 h of unlabeled data and using these for fine-tuning. Attention relaxation is applied in training only, except for "matched inference" (attention relaxation in training *and* test). Best results for each of the three fine-tuning setups are in **bold** font.

**Models and training**    We use `AV-HuBERT` models[3], introduced recently by Shi et al. (2022), which receive image and acoustic frames for pre-training by unlabeled training data to iteratively learn contextualized feature representations $\mathbf{h}_1^T$. For fine-tuning and inference, only the video input is used and preprocessed (cf. preprocessing layer in Figure 2, Appendix A.1) with a 3D convolutional layer and a subsequent `ResNet-18` (He et al., 2016; Stafylakis & Tzimiropoulos, 2017) architecture. The models fine-tuned on 30 h of LRS3 training data use the `base` configuration of the downloaded `AV-HuBERT` encoder and have a total of 160M parameters. Models fine-tuned on 433 h of LRS3 training data (with or without self-training) use the `large` `AV-HuBERT` encoder and comprise 477M parameters in total. As additional regularization methods we use label smoothing (Müller et al., 2020), LayerDrop (Fan et al., 2020), as well as dropout (Srivastava et al., 2014). For final experiments, we use the self-training (Zoph et al., 2020) method, where an `AV-HuBERT` model fine-tuned on 433 h of LRS3 training data is inferred to generate pseudo-labels for the 1,326 h of unlabeled Voxceleb2 data. These were then used together with the true labels from the LRS3 training data to fine-tune the pre-trained `AV-HuBERT` model. Relaxed attention was only used during this final fine-tuning, and relaxation coefficient $\gamma$ of each relaxed attention approach was optimized on the development set for each corresponding amount of fine-tuning data. See Appendix A.3.2 for more details.

---

[3]Pre-trained `AV-HuBERT` models and fine-tuning code downloaded from **https://github.com/facebookresearch/av_hubert**

**Results and discussion** The upper segment of Table 2 shows various baselines on LRS3, whereby Makino et al. (2019) reached 33.60% WER w/o LM, using 33,000 h of YouTube training data, and Ma et al. (2021) achieved 43.30% with LM and 157 h of additional data from the Lip Reading in the Wild dataset (Chung & Zisserman, 2016). By leveraging pre-training of AV-HuBERT models, Shi et al. (2022) report state of the art so far on LRS3 in three cases with 1,326 h unlabeled pre-training data plus 30 h, plus 433 h, plus 433 h + 1,326 h of fine-tuning data, respectively, the latter using self-training to leverage the pre-training data using pseudo-labels. See also our resimulated numbers of that approach. Note that as it is common practice on the LRS3 task to not even report performance on dev condition, we also formulate performance claims on the test set. Smooth focus (Chorowski et al., 2015) helps a bit in 4 out of the 6 total test conditions. Without a language model—adding virtually no parameters and only marginally more complexity during training—our relaxed self-attention achieves WERs of 45.04% vs. 45.90% from Shi et al. (2022), resimulated, and 28.84% vs. 29.52% from Shi et al. (2022), resimulated, in the 30 h and 433 h fine-tuning cases, respectively, with matched inference (relaxation in training and test). With self-training (433 h + 1,326 h), relaxed self-attention without matched inference even achieves **26.31%** WER compared to the best lip-reading WER of 26.90% from Shi et al. (2022) thus setting a new state of the art for LRS3. With an additional LM, similar to the ASR task in Section 4.1, relaxed cross attention yields consistent improvement on the test set compared to the resimulated baseline in all three fine-tuning cases (i.e., **42.68%** vs. 45.33%, **28.05%** vs. 28.97%, and **25.51%** vs. 26.50%, respectively). We show that this is also caused by the improved internal language model handling for this task in Appendix A.4. We also investigate robustness towards different initialization seeds in Appendix A.5.

### 4.3 Application to machine translation

**Task and datasets** Neural machine translation (NMT) models use neural networks that translate an input text sequence from a source language to a different target language. For our particular experiments on relaxed attention we use data from the well-known IWSLT14 translation challenge (Cettolo et al., 2014), choosing the German-to-English (DE→EN) subtask and report performance in terms of BLEU scores (Papineni et al., 2002). For training of an external LM we either use the IWSLT14 target language transcripts (160k utterances) or the MuST-C dataset, the latter contains 47% additional transcripts (236k utterances) from TED talks and is available under the Creative Commons BY–NC–ND 4.0 international license (Cattoni et al., 2021).

**Models and training** For training we use the standard encoder-decoder transformer model from Vaswani et al. (2017) in the base configuration with 36.7M parameters and apply cutoff augmentation, which first randomly masks input positions and feature dimensions of the embedded input tokens and second uses a divergence loss to minimize the difference in predictions for different input masks[4] (Shen et al., 2020). The joint dictionary for source and target language comprises 10k tokens generated with SentencePiece (Kudo & Richardson, 2018) and preprocessed with an embedding layer (cf. preprocessing layer in Figure 2, Appendix A.1). As in the previous tasks, to investigate relaxed attention with LM, we trained two transformer LMs of equal size: One LM trained with IWSLT14 training transcripts and an extended LM trained on the MuST-C dataset, respectively. For both, relaxed cross attention and relaxed self-attention, the relaxation coefficient $\gamma$ has been tuned on the development set. See Appendix A.3.3 for more details.

**Results and discussion** In the upper segment of Table 3, we report BLEU scores for recent transformer-based approaches to NMT, whereof we choose the strong approach from Shen et al. (2020) using cutoff augmentation as a baseline and report a somewhat lower BLEU score of 37.42 in our resimulation. Smooth focus here achieves comparable performance to that baseline with small gains when LMs are used. We observe that a LM trained only with the target language training transcripts of the translation model yields no additional information compared to the internally learned language model and thus does not improve performance for most approaches, even the relaxed cross attention that has been strong (with LM) in previous tasks. However, in case of a strong extended LM trained with additional data, relaxed cross attention (only during training again) yields the best performance of **37.96** BLEU, as it suppresses the internal LM. The best performance for the common case without LM is achieved with our relaxed self-attention approach applied during training *and* test, slightly outperforming the previous state-of-the-art BLEU score without additional training data (37.60, Shen et al. (2020)), with a score of **37.67**, exceeding the resimulated baseline

---

[4]Code available from **https://github.com/dinghanshen/cutoff**

| Approach | *without* LM | *with* LM (training transcripts only) | *with* extended LM (additional data) |
|---|---|---|---|
| | test | test | test |
| Vaswani et al. (2017) | 34.40 | — | — |
| Fan et al. (2020) | 34.50 | — | — |
| Wu et al. (2019) | 35.20 | — | — |
| Wu et al. (2021) | 36.88 | — | — |
| Liang et al. (2021) | 37.25 | — | — |
| Shen et al. (2020) | 37.60 | — | — |
| Baseline (Shen et al. (2020), resim.) | 37.42 | 37.42 | 37.62 |
| + smooth focus (Chorowski et al. (2015)) | 37.42 | 37.52 | 37.67 |
| + relaxed cross attention | 37.56 | 37.53 | **37.96** |
| + matched inference | 37.60 | 37.64 | 37.57 |
| + relaxed self-attention | 37.57 | 37.49 | 37.74 |
| + matched inference | **37.67** | **37.67** | 37.71 |

Table 3: **Neural machine translation** results in terms of BLEU scores on the **IWSLT14** task (DE → EN) using encoder-decoder **transformer models with cutoff augmentation** (Shen et al., 2020). Attention relaxation is applied in training only, except for "matched inference" (attention relaxation in training *and* test). Best results across all approaches are in **bold** font, second best underlined.

even by 0.25 BLEU. We note, that in Iyer et al. (2021) (only available as preprint) the authors also chose the model of Shen et al. (2020) as baseline but were able to reproduce the result of 37.60 BLEU. They report a BLEU score of 37.78 by simply applying a modified learning rate schedule achieving a somewhat smaller improvement of 0.18 BLEU absolute vs. their baseline. Without claiming a new state of the art, we note that both, our and their method are top-performing on the IWSLT14 task. In Appendix A.5, also show robustness of the relaxed self-attention method towards different initialization seeds. Ablations on attention dropout and the sensitiveness of $\gamma$ are shown in Appendices A.6 and A.7.

## 4.4 APPLICATION TO IMAGE CLASSIFICATION

**Task and datasets**   Image classification is a fundamental task in computer vision aiming at recognizing the primary content of images and differs significantly from the previous sequence-to-sequence tasks as it uses a type of an attention-based encoder-only (decoder-less) transformer model, which recently dominate vision benchmarks. To investigate if relaxed attention is also applicable to such tasks, we evaluate performance in terms of classification accuracy (%) on the computationally less demanding CIFAR-100 dataset (A. Krizhevsky, 2009). For each of its 100 classes, it contains 500 and 100 images for training and test, respectively, and is publicly available without a specified license. As initialization, we use a model pre-trained on the ImageNet-1k dataset (Deng et al., 2009), which contains 1.28M training images from 1,000 classes and is also available for research purposes upon agreement of the terms of access.

**Models and training**   For our experiments we use the vanilla `Swin-T` transformer model (Liu et al., 2021b) as baseline—a recently established vision transformer comprising 29M parameters using localized attention. Details on the architecture (including figures) are given in Appendix A.2. For training settings we follow Liu et al. (2021b). For some experiments we downloaded the official ImageNet-1k pre-trained model[5] and report results after fine-tuning for 100 epochs on CIFAR-100 training data. With or without pre-training, relaxed self-attention is applied only during fine-tuning. We investigate the interaction of our relaxed self-attention approach with other regularization methods by omitting already employed (i.e., the well-known stochastic depth method (Huang et al., 2016)) or adding recently proposed (i.e., the dense relative localization loss $\mathcal{L}_{\mathrm{drloc}}$ (Liu et al., 2021a)) approaches. For fair comparison and following common practice as in (Kwon et al., 2021; Liu et al., 2021a; Wang et al., 2017), we report results of our relaxed self-attention approaches after roughly optimizing `test` accuracy with a small grid search over $\gamma$ values (and $\sigma^2$ for fuzzy relaxation after the optimal $\gamma_0$ was found) separately for each batch size (1024 and 128) with pre-training, applying the found values to experiments without pre-training. See Appendix A.3.4 for more details.

---

[5]ImageNet-1k pre-trained `Swin` transformer models and fine-tuning code downloaded from `https://github.com/microsoft/Swin-Transformer.`

| Approach | | w/ pre-training batch size | | w/o pre-training batch size | |
|---|---|---|---|---|---|
| | | 1024 | 128 | 1024 | 128 |
| | | test | test | test | test |
| Other transformers | Dosovitskiy et al. (2021), `ViT-S-16` | 87.10 | — | — | — |
| | Yuan et al. (2021), `T2T-ViT-14` | 88.40 | — | — | — |
| | Guo et al. (2021), `CMT-S` | 91.70 | — | — | — |
| Swin-T transformers | Liu et al. (2021b), `Swin-T` (vanilla) | 88.22 | — | 53.28 | — |
| | Liu et al. (2021a), `Swin-T` (+ $\mathcal{L}_{\mathrm{drloc}}$) | 88.40 | — | **66.23** | — |
| | Baseline (Liu et al. (2021b), resimulated) | 88.53 | 89.16 | 53.12 | 64.10 |
| | - stochastic depth (Huang et al. (2016)) | 87.62 | 88.42 | 57.62 | **68.08** |
| | + $\mathcal{L}_{\mathrm{drloc}}$ (Liu et al. (2021a)) | 88.63 | 89.27 | 61.72 | 66.29 |
| | + smooth focus (Chorowski et al. (2015)) | 88.44 | 89.53 | 57.02 | 64.15 |
| | + relaxed self-attention | 88.64 | 89.21 | 52.89 | 63.72 |
| | + matched inference | **88.73** | 89.39 | 53.15 | 63.52 |
| | - stochastic depth (Huang et al. (2016)) | 87.49 | 88.42 | 56.99 | 67.91 |
| | + $\mathcal{L}_{\mathrm{drloc}}$ (Liu et al. (2021a)) | 88.55 | 89.29 | 61.37 | 65.90 |
| | + fuzzy relaxation | 88.63 | **89.60** | 52.51 | 63.58 |

Table 4: **Image classification** results in terms of accuracy (%) on the **CIFAR-100** task using encoder-only **transformer** models. Relaxed self-attention is applied in training only, except for "matched inference" (relaxation in training *and* test). All reference methods have roughly the same model size and complexity. Best results across all `Swin-T` approaches are in **bold** font, second best underlined.

**Results and discussion** The first segment of Table 4 shows results for reference vision transformer models ranging from 87.10% accuracy for the pure attention-based `ViT-S-16` (Dosovitskiy et al., 2021) to 91.70% accuracy for the convolution attention hybrid model `CMT-S` (Guo et al., 2021). The second table segment presents baselines and experimental results for `Swin-T` transformer models where we chose the vanilla architecture (Liu et al., 2021b) to resimulate a baseline for our experiments. Omitting stochastic depth (Huang et al., 2016) causes a severe loss of performance with pre-training but clearly helps when training from scratch. For the dense relative localization loss $\mathcal{L}_{\mathrm{drloc}}$ (Liu et al., 2021a), we confirm performance gains with and especially without pre-training. Smooth focus helps for the small batch size using pre-training and performs remarkably good for a large batch size when training from scratch. Without pre-training we observe that relaxed self-attention doesn't help. This might be due to the limited number of training epochs and a slower convergence caused by the additional relaxed self-attention regularization, similar to the effect of stochastic depth in the resimulated baseline. When applying relaxed attention after pre-training, however, relaxed self-attention alone slightly outperforms the baseline but achieves even higher accuracies when used with matched inference (**88.73**% vs. 88.53%) and (89.39% vs. 89.16%) for the large and small batch sizes, respectively. Matched inference turned out to be advantageous on this task in most cases, thus we continue to report based thereon. Also, we note that the combination with stochastic depth seems to be beneficial for relaxed self-attention. Our new fuzzy relaxation with matched inference turns out to be useful only on smaller batch sizes after pre-training, achieving a strong accuracy of **89.60**% outperforming the baseline (Liu et al. (2021b), resimulated) at 89.16%. We also investigate robustness towards different initialization seeds in Appendix A.5.

## 5 CONCLUSIONS

In this work we broadly explored the idea of relaxed attention for transformer architectures, a simple smoothing method of the attention weights in the attention layers. We confirmed the advantage of relaxed cross attention when combined with strong external language models and introduced relaxed self-attention, thereby providing regularization also in the transformer encoder and increasing the versatility of relaxed attention to different transformer variants. We show improvements when applying relaxed attention to automatic speech recognition, lip-reading, machine translation, and image classification. On the LRS3 lip-reading task in particular we achieve a word error rate of **26.31%** (vs. the former state of the art of 26.90%) as well as a top-performing BLEU score of **37.67** on the IWSLT14 machine translation task.

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

# A   APPENDIX

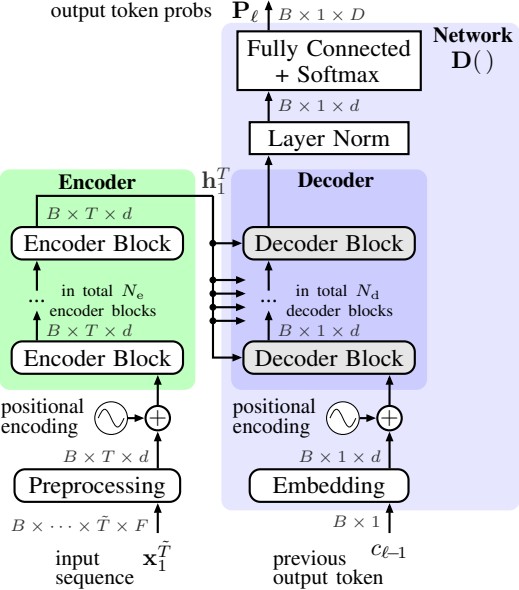

Figure 2: Standard **encoder-decoder transformer** during inference as used for sequence-to-sequence tasks (i.e., automatic speech recognition, lip-reading, and machine translation).

## A.1   MODEL SPECIFICS FOR THE SEQUENCE-TO-SEQUENCE TRANSFORMER

In this section we briefly review the original transformer architecture from Vaswani et al. (2017) consisting of encoder *and* decoder as shown in Figure 2. Please note that here we describe the transformer architecture exactly as used for the investigated sequence-to-sequence tasks (i.e., automatic speech recognition, lip-reading, and machine translation), employing task-dependent individual preprocessing steps, while the encoder-only `Swin` transformer, used for the image classification task, is separately described and shown in Appendix A.2.

### A.1.1   ENCODER-DECODER TRANSFORMER

The input sequence $\mathbf{x}_1^{\tilde{T}}$ of length $\tilde{T}$ (and more optional dimensions, e.g., for lip-reading: video channel, height, and width, or for ASR: acoustic feature dimension $F$) is entirely fed to the transformer's encoder and auto-regressively transformed (by the decoder model $\mathbf{D}(\,)$) into an output token sequence $c_1^L = (c_1, c_2, \ldots, c_L)$ with $c_\ell \in \mathcal{C} = \{c^{(1)}, c^{(2)}, \ldots, c^{(D)}\}$ being a single output token (i.e., grapheme-based characters or (sub-) word units (Kudo & Richardson, 2018)) at output sequence index $\ell \in \{1, \ldots, L\}$ from a vocabulary of size $D$. Specifically, the original input sequence $\mathbf{x}_1^{\tilde{T}}$ is first subject to a task-dependent preprocessing that outputs a feature sequence of $t \in \{1, \ldots, T\}$ frames, optionally sub-sampled with $T \le \tilde{T}$. For each decoding step (starting at $\ell = 1$), the transformer decoder uses the entire encoded input sequence $\mathbf{h}_1^T$ and the previous output token $c_{\ell-1}$ to finally output a vector $\mathbf{P}_\ell = \mathbf{D}(\mathbf{h}_1^T, c_{\ell-1})$ comprising probabilities of all $D$ possible output tokens. These probabilities are then subject to a beam search algorithm which, step-by-step, invokes the decoder until an end-of-sentence (EOS) threshold is exceeded and the final set of hypotheses is emitted. Considering regularization, the standard encoder-decoder transformer model employs three different variants of dropout (Srivastava et al., 2014): Residual dropout applied to sub-layer outputs before the residual connection is added, activation dropout applied after the rectified linear unit (ReLU) activation, and attention dropout which is applied to the attention weights inside the MHA function (all layers, where dropout might be applied to the respective outputs, are shown as dashed boxes in Figures 1 and 3).

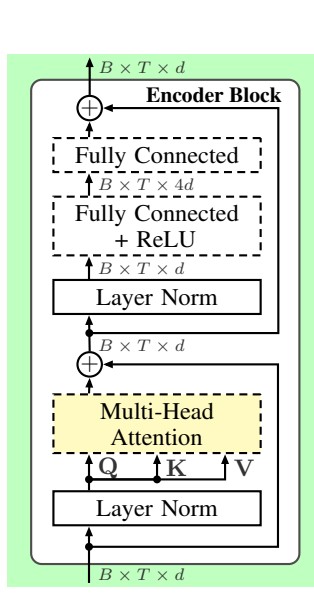

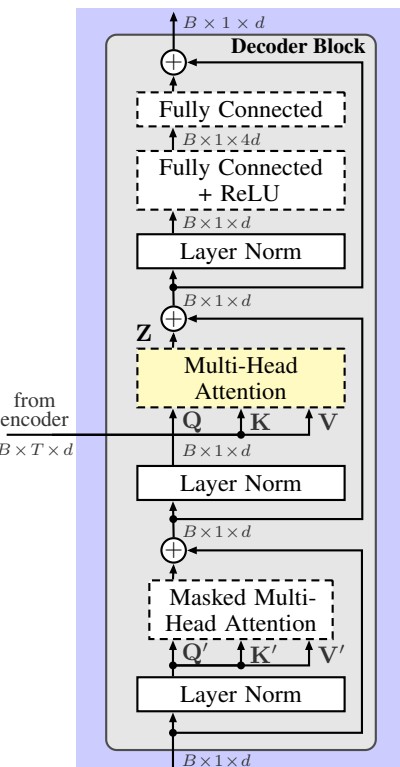

(a) Single **encoder block** with self-attention.     (b) Single **decoder block** with cross attention.

Figure 3: **Encoder and decoder blocks** as used in the transformer model (Figure 2) during inference. Multi-head attention blocks *which may exhibit relaxed attention* are colored yellow. Details thereof are shown in Figure 1. Layers, where dropout (Srivastava et al., 2014) might be applied to the outputs, are depicted as dashed-line boxes.

### A.1.2 SCALED DOT-PRODUCT ATTENTION

Besides other variants of the original attention function introduced in Bahdanau et al. (2015), in this work we focus on scaled dot-product multi-head attention (MHA), introduced together with the orignal encoder-decoder transformer model (Vaswani et al., 2017). As shown in Figure 1 (without the red block), the standard MHA employs multiple (i.e., $N_\mathrm{h}$) attention heads

$$\mathbf{Z}_i(\mathbf{Q},\mathbf{K},\mathbf{V}) = \mathbf{softmax}\underbrace{\left(\frac{\mathbf{Q}\mathbf{W}_i^{(\mathrm{Q})}\left(\mathbf{K}\mathbf{W}_i^{(\mathrm{K})}\right)^\mathsf{T}}{\sqrt{d}}\right)}_{\substack{\text{attention weights}\\=\mathbf{G}_i(\mathbf{Q},\mathbf{K})}} \cdot \underbrace{\mathbf{V}\mathbf{W}_i^{(\mathrm{V})}}_{\substack{\text{value projections}\\=\mathbf{Y}_i(\mathbf{V})}} \in \mathbb{R}^{\tilde{L}\times\frac{d}{N_\mathrm{h}}} \tag{2}$$

with $\mathbf{W}_i^{(\mathrm{Q})}, \mathbf{W}_i^{(\mathrm{K})}, \mathbf{W}_i^{(\mathrm{V})} \in \mathbb{R}^{d\times\frac{d}{N_\mathrm{h}}}$ being linear projection weight matrices for the query $\mathbf{Q}$, key $\mathbf{K}$, and value $\mathbf{V}$ inputs, $i \in \mathcal{N}_\mathrm{h} = \{1\ldots N_\mathrm{h}\}$ being the index of the in total $N_\mathrm{h}$ attention heads, and $d$ is the feature vector size being used in most layers of the transformer model often referred to as the model dimension. Considering cross attention, key and value inputs stem from the encoder's last layer, yielding $\mathbf{K}=\mathbf{V}=\mathbf{h}_1^T$ and the entries in each of the $\tilde{L}=L$ rows of the attention weight matrix $\mathbf{G}_i(\mathbf{Q},\mathbf{K}) \in \mathbb{I}^{\tilde{L}\times T}$, with $\mathbb{I}=[0,1]$, sum up to one and are treated as probabilities that correspond to the relevance of a time frame $t$ to the $\ell$-th or $t$-th position in the query input for cross attention or self-attention, respectively. The outputs $\mathbf{Z}_i$ of all $N_\mathrm{h}$ separate attention heads are concatenated and subject to a fully connected output layer, yielding the MHA output $\mathbf{Z} \in \mathbb{R}^{\tilde{L}\times d}$. Note that for brevity of notation the attention dropout commonly applied to the attention weights in transformer models is not shown in equation (2).

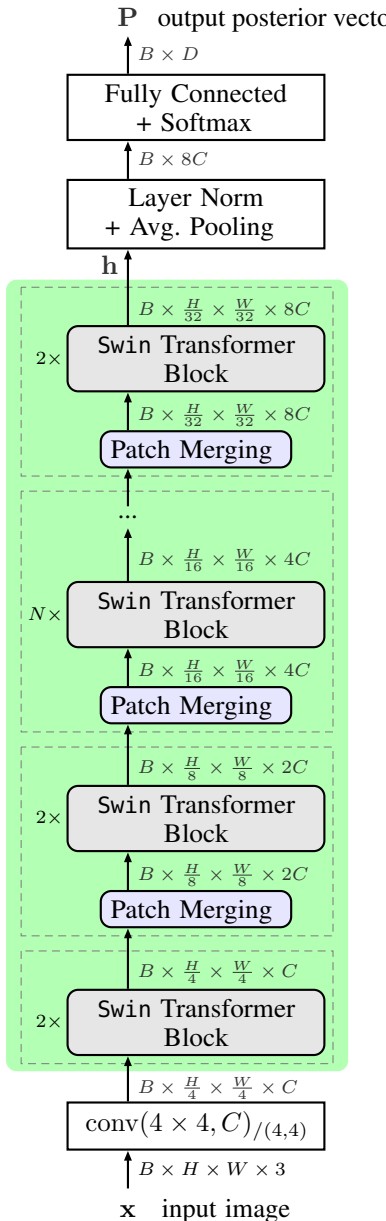

Figure 4: **Swin transformer** as used for the image classification tasks.

## A.2  MODEL SPECIFICS FOR THE VISION TRANSFORMER

As attention-based model for the image classification task we employ the recently successful encoder-only transformer, dubbed the `Swin` transformer (Liu et al., 2021b), as shown in Figure 4. The `Swin` transformer is a hierarchical vision transformer, which uses a shifting window scheme to compute its feature representations $\mathbf{h}$ and can be used as a general purpose backbone for various vision tasks. In contrast to the sequence-based tasks, where a whole decoder is employed to yield sequential output, here, a single fully connected layer with softmax activation (and preceding layer normalization and adaptive average pooling) is used after the `Swin` transformer blocks to assign probabilities $\mathbf{P}$ to the $D$ classes for each image inside a batch $B$.

As input, the `Swin` transformer receives an RGB input image $\mathbf{x} \in \mathbb{I}^{H \times W \times 3}, \mathbb{I} = [0, 1]$ of height $H$ and width $W$, which is divided into non-overlapping image patches of size $4 \times 4$ by a convolutional

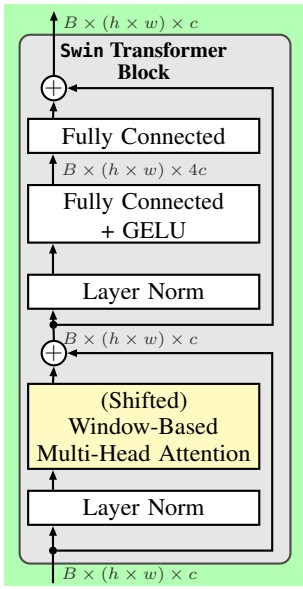

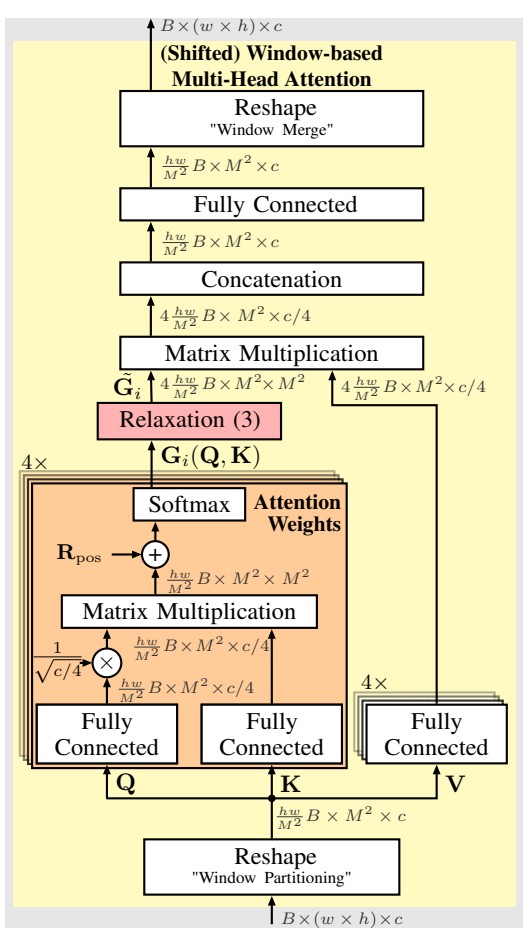

(a) Single **Swin transformer block** as used in the Swin transformer model (cf. Figure 4) during training and inference. Instead of dropout in the encoder-decoder transformer, here we apply stochastic-depth (Huang et al., 2016) to randomly drop layers parallel to the residual connections during training. (Shifted) window-based multi-head attention blocks *which may exhibit relaxed attention* are colored yellow. Details thereof are shown in Figure 5b.

(b) **(Shifted) window-based multi-head attention (MHA)** as used in the Swin transformer block Figure 5a with $N_{\mathrm{h}} = 4$ attention heads. The proposed **relaxed attention** (red block) is presented in Section A.2.

Figure 5: Details of a **Swin transformer block** and the **(shifted) window-based multi-head attention (MHA)**, where relaxed attention (red block) is applied for the image classification task.

layer with a stride of $(4, 4)$ and is thereby embedded into a feature representation of dimension $C$. The hierarchical structure of the Swin transformer consists then of four stages each depicted as a dashed box in Figure 4. In each stage, the patch merging modules first reduce the spatial resolution and double the feature dimensionality ($n \cdot C \rightarrow 2n \cdot C$), while dimensions remain constant for the subsequent processing of that specific stage through the specified number of Swin transformer blocks for that specific stage.

The Swin transformer block, shown in Figure 5a, is based on the original standard transformer block (Vaswani et al. (2017), see also Figure 3a), but replaces the ReLU activation with a Gaussian error linear unit (GELU) activation function after the first fully connected layer and, more importantly, uses a (shifted) window-based multi-head attention module, shown in Figure 5b. There, the window partitioning limits the self-attention computation to non-overlapping local $M \times M$ windows ($M = 7$), which are shifted in position every other Swin transformer block. Once the features are split into windows, they are treated as separate batch instances yielding a temporary batch size of $\frac{hw}{M^2}B$ with $B$ being the original batch size. Different to the standard multi-head attention, a relative position bias $\mathbf{R}_{\mathrm{pos}} \in \mathbb{R}^{M^2 \times M^2}$ is added before softmax activation. The attention weights $\mathbf{G}_i(\mathbf{Q}, \mathbf{K}) \in \mathbb{R}^{M^2 \times M^2}$

inside the shifted window-based MHA contain probabilities for relevant entries in these windows and are then subject to the herein investigated relaxation (see red box in Figure 5b), yielding

$$\tilde{\mathbf{G}}_i(\mathbf{Q}, \mathbf{K}) = \left[ (1-\gamma)\mathbf{G}_i(\mathbf{Q}, \mathbf{K}) + \gamma \frac{\mathbf{1}}{M^2} \right], \ \ i \in \mathcal{N}_{\mathrm{h}}, \tag{3}$$

with $M^2$ being the fixed amount of features in a single window. See Section 3 for the sequence-based relaxed attention approach as well as for the fuzzy relaxation which randomly varies the relaxation coefficient $\gamma$ to compensate for the now constant $M^2$ term in equation (3).

### A.3 EXPERIMENTAL DETAILS

In this section we will list additional experimental details for all of the investigated tasks, thereby starting with general settings that apply to multiple tasks and then providing details for the experiments of each individual task that has been investigated in this work. Please note that for all tasks we used publicly available code as baselines and did not change any hyper-parameters unless explicitly mentioned (e.g., for ablation studies).

In experiments where an additional language model was included, we used the common shallow fusion method (Gülçehre et al., 2015) for language model fusion. Specifically, shallow fusion combines the output token probability vector $\mathbf{P}_\ell$ at the output of the transformer model (cf. Figure 2) for each decoding timestep $\ell$ with the same $D$-length output token probabilities $\mathbf{P}_\ell^{(\mathrm{LM})}$ in the logarithmic domain to gather a joint output token probability $\log \tilde{\mathbf{P}}_\ell = \log \mathbf{P}_\ell + \lambda \log \mathbf{P}_\ell^{(\mathrm{LM})}$. The language model weight is used to steer the influence of the language model during decoding and is gathered individually for each task.

In all investigated tasks, the smooth focus method from Chorowski et al. (2015) is applied as a reference method that smoothes the attention weights in the cross attention layer by modyifing the softmax function. Defining the scaled dot-product of query and key projections, which is input to the softmax function in equation (2), as $\mathbf{E}_i = \frac{1}{\sqrt{d}} \mathbf{Q} \mathbf{W}_i^{(\mathrm{Q})} \left( \mathbf{K} \mathbf{W}_i^{(\mathrm{K})} \right)^{\mathsf{T}} = (e_{i,\ell,t}) \in \mathbb{R}^{L \times T}$ with $e_{i,\ell,t}$ being elements thereof, the single elements $\mathrm{g}_{i,\ell,t}$ of the attention weights $\mathbf{G}_i(\mathbf{Q}, \mathbf{K})$ with smooth focus are computed as

$$\mathrm{g}_{i,\ell,t}(\mathbf{Q}, \mathbf{K}) = \frac{\sigma_{\mathrm{sig}}\left(e_{i,\ell,t}\right)}{\sum_{t=1}^{T} \sigma_{\mathrm{sig}}\left(e_{i,\ell,t}\right)}, \tag{4}$$

with $\sigma_{\mathrm{sig}}$ being the sigmoid function, which for smooth focus replaces the unbounded exponential function from the standard softmax function. Please note that for the Swin transformer the softmax input can be defined analogously as $\mathbf{E}_i = \mathbf{R}_{\mathrm{pos}} + \frac{1}{\sqrt{c/4}} \mathbf{Q} \mathbf{W}_i^{(\mathrm{Q})} \left( \mathbf{K} \mathbf{W}_i^{(\mathrm{K})} \right)^{\mathsf{T}} \in \mathbb{R}^{M^2 \times M^2}$.

#### A.3.1 AUTOMATIC SPEECH RECOGNITION

The specific model architecture for the trainings with 100 h and 960 h of training data, are standard encoder-decoder transformer models with a small ($N_{\mathrm{e}} = N_{\mathrm{d}} = 6, N_{\mathrm{h}} = 4, d = 512$) and a large ($N_{\mathrm{e}} = 12, N_{\mathrm{d}} = 6, N_{\mathrm{h}} = 4, d = 512$) configuration, respectively, with $N_{\mathrm{e}}$, $N_{\mathrm{d}}$, and $N_{\mathrm{h}}$ being the number of encoder blocks, decoder blocks, and attention heads, respectively, and $d$ is the model dimension (i.e., the amount of nodes used in most layers of the model). The external RNN language model consists of shared input and output embedding layers with four LSTM layer in between each comprising 800 nodes yielding a total of 24.5M parameters.

Training of ASR models was done using the `espresso` toolkit, which is an extension of the `PyTorch`-based `fairseq` toolkit. We performed a small grid search on the joint `dev clean` and `dev other` datasets among values $\{0.0001, 0.001, 0.01, 0.05, 0.1\}$ and $\{0.1, 0.15, 0.2, 0.25, 0.3\}$ for the relaxation coefficient $\gamma$ and found $\gamma = 0.01$ and $\gamma = 0.25$ to be optimal for relaxed self-attention and relaxed cross attention, respectively. Optimal values were used for both, 100 h and 960 h training data configurations. All remaining training hyper-parameters were adopted from the recipes available at `https://github.com/freewym/espresso` with commit id `390ad6f`. Specifically, we train all models for 100 epochs using the Adam optimizer (Kingma & Ba, 2015) with a learning rate of $0.001$. All dropout layers (i.e., residual, activation, and attention dropout) used the dropout rate $p = 0.2$ and

the label smoothing coefficient was set to $\alpha = 0.1$. Models for 100 h of training data were trained using a single `RTX2080ti` GPU, while larger models on 960 h of training data were trained on a single `A100` GPU.

Inference was done using a beam search with beam size of 60 and the language model weight $\lambda$ was fixed at $0.4$, following recipes from Wang et al. (2019) for all experiments with LM, without further optimization.

### A.3.2 LIP-READING

The specific model architecture for fine-tuning with 30 h of labeled data, is a pre-trained `base AV-HuBERT` encoder model with an appended standard transformer decoder model ($N_e\!=\!12, N_d\!=\!6, N_h\!=\!12, d\!=\!768$) while for the 433 h and 433 h + 1,326 h setups a `large AV-HuBERT` encoder with a larger decoder was used ($N_e\!=\!24, N_d\!=\!9, N_h\!=\!16, d\!=\!1024$). The external transformer language model comprises 16 decoder blocks with $d = 512$ (cf. Figure 3b, but without the cross attention layer) and uses a shared input/output embedding of the in total $D = 1000$ subword units, resulting in a language model size of 51M parameters.

Training of lip-reading models was done using the `PyTorch`-based `fairseq` toolkit. We performed a small grid search on the development dataset among values $\{0.005, 0.01, 0.02, 0.05, 0.1\}$ and $\{0.1, 0.15, 0.2, 0.25, 0.3\}$ for the relaxation coefficient of relaxed self-attention and relaxed cross attention, respectively. For the 30 h fine-tuning case we found $\gamma = 0.001$ and $\gamma = 0.25$, for 433 h we found $\gamma = 0.005$ and $\gamma = 0.25$, and for the 433 h+1,326 h case we found $\gamma = 0.005$ and $\gamma = 0.2$ to be optimal. All remaining training hyper-parameters were adopted from the recipes available at `https://github.com/facebookresearch/av_hubert` with commit id `cd1fd24`. Residual, activation, and attention dropout layers were using a dropout rate $p$ of 0.1, 0.1, and 0.0, respectively. The label smoothing coefficient was set to $\alpha = 0.1$. Models for the smaller 30 h fine-tuning data setup were trained using a single `RTX3080` GPU, while for all other settings a single `A100` GPU was used for training.

Inference was done using a beam search with beam size of 50 and the language model weight $\lambda$ was optimzied for each approach by searching optimal values on the development data among values of $\{0.05, 0.1, 0.15, 0.2\}$.

### A.3.3 MACHINE TRANSLATION

The standard encoder-decoder transformer from Vaswani et al. (2017) was used in the `base` configuration ($N_e\!=\!N_d\!=\!6, N_h\!=\!4, d\!=\!512$). The external transformer language model consists of shared input and output embedding layers of the in total $D = 10000$ subword units with 6 decoder blocks (cf. Figure 3b, but without the cross attention layer) in between and comprises 24.1M parameters.

Training of the machine translation transformer models was done using the `PyTorch`-based `fairseq` toolkit. We performed a small grid search on the development dataset among values $\{0.005, 0.01, 0.02, 0.05, 0.1\}$ and $\{0.1, 0.15, 0.2, 0.25, 0.3\}$ for the relaxation coefficient and found $\gamma = 0.05$ and $\gamma = 0.1$ optimal for relaxed self-attention and relaxed cross attention, respectively. For approaches with LM the language model weight $\lambda$ was tuned among values $\{0, 0.05, 0.1, 0.15, 0.2\}$. All remaining training hyper-parameters were adopted from the recipes available at `https://github.com/dinghanshen/Cutoff` with commit id `4978563`. Residual, activation, and attention dropout layers were set to $0.3$, $0.1$, and $0.1$, respectively. All models were trained using a single `RTX2080ti` GPU.

Inference was done using a beam search with beam size of 10 and the language model weight $\lambda$ was optimized for each approach by searching optimal values on the development dataset among values of $\{0.05, 0.1, 0.15, 0.2\}$.

### A.3.4 IMAGE CLASSIFICATION

We chose the `Swin` transformer as the specific model architecture for trainings with and without pre-training. It is a multi-purpose backbone for various vision tasks and can be configured in terms of size and complexity. Specifically, we use the tiny configuration of the model dubbed `Swin-T`, which is defined by an initial feature embedding dimensionality $C = 96$ and comprises $N = 6$

`Swin` transformer blocks in the third stage, resulting in a total of $N_e = 12$ `Swin` transformer blocks. The number of attention heads $N_h$ doubles with each consecutive stage, yielding an amount of $\{3, 6, 12, 24\}$ attention heads for the respective stages. In total, the model comprises 29M parameters.

Training of image classification models was done using the `PyTorch` toolkit. We performed a small grid search among values $\{0.005, 0.01, 0.05, 0.1, 0.15, 0.2\}$ and $\{0.01, 0.02, 0.03\}$ for the relaxation coefficient of relaxed self-attention and $\sigma^2$ for fuzzy relaxation, respectively. Following common practice on the CIFAR-100 task (see, e.g., Kwon et al. (2021); Liu et al. (2021a); Wang et al. (2017)), parameter search was conducted on the `test` dataset. For the training with pre-training we found $\gamma = 0.1$ and $\sigma = 0.03$ to be optimal. Both found relaxation hyper-parameters were also applied for experiments without pre-training. All remaining training hyper-parameters were adopted from the recipes available at `https://github.com/microsoft/Swin-Transformer` with commit id `5d2aede`. For some trainings we use an auxiliary dense relative localization loss $\mathcal{L}_{\mathrm{drloc}}$, which encourages vision transformers to learn spatial information between image patches and thereby boosts convergence, especially for small datasets (Liu et al., 2021a). For the $\mathcal{L}_{\mathrm{drloc}}$ loss, we adopted the official `Swin`-based code from `https://github.com/yhlleo/VTs-Drloc` with commit id `b69adb6`. Specifically we train all models for 100 epochs using the Adam optimizer (Kingma & Ba, 2015) with a learning rate of 0.000125 for a batch size of 128 and 0.001 for a batch size of 1024. Stochastic depth (Huang et al., 2016), which randomly drops layers in the transformer block, is a standard method for training the baseline model and was used with a drop probability of 0.2. Label smoothing was used with a value of 0.1. All `Swin` transformer models were trained using a single `RTX2080ti` GPU.

## A.4 INTERNAL LANGUAGE MODEL SUPPRESSION

As shown in Table 1 for automatic speech recognition, we achieved superior results with relaxed cross attention *only* when the transformer was combined with an external language model that is trained with large amounts of additional text-only data. This finding is in line with Lohrenz et al. (2021), but Lohrenz et al. (2021) does not provide a sound reason for such behavior. Different to hybrid ASR approaches, the output token posterior $\mathbf{P}_\ell$ of a trained transformer model cannot technically be decomposed into an acoustic model $\mathrm{P}(\mathbf{x}_1^T | c_1^L)$ and language model $\mathrm{P}(c_1^L)$, since the latter is also implicitly learned on the training transcripts by the transformer decoder that in addition to the encoder output autoregressively receives previous output tokens as it is the case for language models.

| Approach | absolute WER | | | | LM-induced WER reduction | | | |
| | dev | | test | | dev | | test | |
| | clean | other | clean | other | clean | other | clean | other |
| --- | --- | --- | --- | --- | --- | --- | --- | --- |
| Baseline (Lohrenz et al. (2021), resim.), no LM | 3.92 | 9.00 | 4.47 | 9.23 | — | — | — | — |
| + LM (training transcripts only) | 3.92 | 8.90 | 4.44 | 9.20 | 0.00 | 0.10 | 0.03 | 0.03 |
| + LM (additional data, from Tab. 1) | 3.73 | 8.52 | 4.40 | 8.95 | 0.19 | 0.48 | 0.07 | 0.28 |
| Relaxed cross attention, no LM | 3.95 | 9.33 | 4.28 | 9.45 | — | — | — | — |
| + LM (training transcripts only) | 3.91 | 9.26 | 4.23 | 9.30 | 0.04 | 0.07 | 0.05 | 0.15 |
| + LM (additional data, from Tab. 1) | **3.44** | **7.74** | **3.58** | **8.35** | 0.51 | 1.59 | 0.70 | 1.10 |

Table 5: **Automatic speech recognition** results in terms of WER (%) on the **Librispeech** task using standard **encoder-decoder transformer** models. The 960 h training dataset is used, see also Table 1.

Here, we investigate whether the improvement by relaxed cross attention might be due to a suppression of the *internal language model*. To accomplish this, in Table 5, we measure the WER improvement achieved by using an LM when the transformer was trained with and without relaxed cross attention, respectively. Both trained transformer models are combined with two language models, one trained *only* from the text transcripts of the acoustic training data, and one trained with additional text-only data. Note that both, resimulated baseline results and results for the LM with additional data, are taken from Table 1. We observe that for both, the baseline and the relaxed cross attention model, the improvements with the *training transcript only* LM (rows 2 and 5) vs. the no LM methods are about equally small. In contrast, when combined with the LM model trained on additional data, the model trained with relaxed cross attention yields far more WER reduction as if this strong LM would be used with the baseline. In any case it exceeds an absolute reduction of 0.5% (nowhere reached

with the baseline), and for the (noisy) `other` condition it is more than 1% absolute WER reduction if relaxed cross attention is employed.

| Approach | absolute WER | | LM-induced WER reduction | |
|---|---|---|---|---|
| | dev | test | dev | test |
| Baseline (Shi et al. (2022), resimulated), no LM | 17.71 | 26.73 | — | — |
| + LM (training transcripts only) | 17.83 | 27.22 | -0.12 | -0.49 |
| + LM (additional data, from Tab. 2) | 17.18 | 26.50 | 0.53 | 0.23 |
| Relaxed cross attention, no LM | 17.40 | 26.57 | — | — |
| + LM (training transcripts only) | 17.48 | 26.52 | -0.08 | 0.05 |
| + LM (additional data, from Tab. 2) | **16.92** | **25.51** | 0.48 | 1.01 |

Table 6: **Automatic lip-reading** results in terms of WER (%) on the **LRS3** task using standard **encoder-decoder transformer** models with pre-trained **AV-HuBERT** encoders. For fine-tuning 433 h + 1,326 h of labeled data are used, see also Table 2.

For the automatic lip-reading task we observe similar behavior in Table 6. Here the integration of the training transcripts only LM is even harmful for the baseline model (row 2), while for the relaxed cross attention approach, WERs remain roughly the same compared to the relaxed cross attention-trained model without LM (row 4 vs. 5). In combination with the strong LM, both baseline and relaxed cross attention models take profit on the `dev` set, while on the `test` set, relaxed cross attention yields a more than four-fold WER reduction by LM fusion (1.01% absolute) compared to the baseline approach (0.23% absolute).

Overall, we observe that relaxed cross attention does not yet help when the LM was trained only with the text transcript data that was already exposed to the ASR transformer model during training of acoustic data. We conclude, however, that relaxed cross attention particularly helps when the LM has been trained with additional text data and seems to suppress the internal model bias, thus suppressing the influence of the internally learned (usually poor) language model. Note that the same behavior is observed in Table 3 for neural machine translation.

## A.5    ROBUSTNESS TO DIFFERENT INITIALIZATION SEEDS

In Table 7, we investigate the influence of different initialization seeds on our experiments. While for the main experiments in Section 4 we experimented on an unchanged and non-optimized seed for random number generation, here—since both of our SOTA contributions are based on the novel self-attention—we analyze the best relaxed self-attention schemes of each task w.r.t. statistical significance when using 5 different random seeds.

| Task | *Automatic Speech Recognition* (Librispeech) | | *Automatic Lip-Reading* (LRS3) | *Machine Translation* (IWSLT14) | *Image Classification* (CIFAR-100) |
|---|---|---|---|---|---|
| Setting | 100 h training data w/o LM | | 433 h + 1,326 h labeled data w/o LM | w/o LM, matched inference | w/ pre-training, batchsize 128 fuzzy relaxation |
| Metric | WER (%) | | WER (%) | BLEU | Acc. (%) |
| Data subset | test clean | test other | test | test | test |
| Baseline (resimulated) | 14.89±0.17 | 29.66±0.34 | 26.92±0.21 | 37.49±0.10 | 89.29±0.12 |
| Relaxed self-attention | **14.25**±0.29 | **28.63**±0.54 | **26.36**±0.22 | **37.66**±0.02 | **89.45**±0.17 |

Table 7: Sensitivity to different initialization of the various tasks. Training of the models for the baseline and **best relaxed self-attention approach** was repeated 5 times. Results are shown in terms of average and standard deviation values of the respective metrics.

We note that in these experiments, *we achieve significant improvement for all three sequence-based tasks including those where we claim state-of-the-art and top performance (i.e., lip-reading and*

*machine translation).* In addition, not shown here, the relaxed cross attention method yielded even better performance on all three sequence-based tasks, outperforming relaxed self-attention, but we do not formulate performance claims in this particular analysis as it implies extra computational complexity due to the requirement of a language model as well as additional unpaired text training data. For the image classification task, note that we reach a clear improvement using the non-optimized standard seed for initialization of our main experiments (see Table 4). Here, however, with additional seeds for initialization, we observe the baseline and the fuzzy relaxation approach to differ without statistical significance. We suspect this is due to non-deterministic operations in the original baseline code from Liu et al. (2021b), which might have flawed the tuning process for the relaxation coefficients for fuzzy relaxation. However, as the average accuracy with fuzzy relaxation is still higher (89.45% vs. 89.29%), we feel encouraged to further expand the relaxed self-attention approach to attention-based approaches for computer vision tasks.

## A.6 ABLATION STUDY ON ATTENTION DROPOUT

Depicted as dashed boxes in Figures 1 and 3, the well-known dropout method (Srivastava et al., 2014) is employed to the standard encoder-decoder transformer in three different variations: Residual dropout, activation dropout, and—most relevant for our study—attention dropout, where the latter is either applied to the attention weights $\mathbf{G}_i$ after the softmax layer (baseline) or to the modified attention weights $\tilde{\mathbf{G}}_i$ after the relaxation operation (relaxed attention approaches, see equation (1)). In Table 8, we investigate how these regularization operations interfere with each other for two different tasks that incorporate attention dropout during training. Therefore, in this ablation, we removed attention dropout throughout the encoder and the decoder of the transformer model for both approaches with and without specified types of relaxed attention. Note that the employed models for the lip-reading and image recognition tasks did not use attention dropout (following the respective baseline recipes from Shi et al. (2022) and Liu et al. (2021b), see experimental details in Appendices A.3.2 and A.3.4) and are thus omitted for this ablation. Specific values for attention dropout are given for each task in Appendix A.3.

| Task | Automatic Speech Recognition (Librispeech) | | | | Machine Translation (IWSLT14) |
|---|---|---|---|---|---|
| Setting | w/ LM 100 h training data | | | | w/o LM, |
| Relaxation type (*) | cross attention | | | | self-attention matched inference |
| Metric | WER (%) ↓ | | | | BLEU ↑ |
| Data subset | dev clean | dev other | test clean | test other | test |
| Baseline (resimulated) | 10.62 | 24.19 | 12.06 | 25.56 | 37.42 |
| - attention dropout | 11.02 | 25.24 | 11.89 | 26.88 | 37.51 |
| + relaxed (*) attention | **9.33** | 22.16 | **10.62** | **23.04** | **37.67** |
|   - attention dropout | 9.68 | **21.38** | 10.91 | 23.16 | 37.47 |

Table 8: Ablation study on attention dropout (Srivastava et al., 2014) for exemplary **automatic speech recognition** and **neural machine translation** tasks. Best results across approaches are in **bold** font and arrows (↓↑) point to the direction of better metric values for each task.

We note that relaxed attention nicely combines with attention dropout (Srivastava et al., 2014) as in the `test` conditions of both tasks the combination of relaxed self-/cross attention with attention dropout yields the best results, which are also reported in the main experiments for both specific tasks in Section 4. Interestingly, attention dropout did even harm the baseline performance for machine translation, as omitting it yields an 0.09 absolute increase in BLEU score, while it improves the advantageous relaxed self-attention even further. In summary, we observe that both proposed relaxed attention approaches seem to go along with other regularization approaches, such as attention dropout, providing complementary regularization to the attention layers.

## A.7 SENSITIVENESS OF RELAXATION COEFFICIENT $\gamma$

In Figures 6 and 7 we investigate the sensitiveness of the relaxation coefficient $\gamma$ for the automatic speech recognition and for the neural machine translation task respectively. As introduced in Section 3 the constant relaxation coefficient $\gamma \in [0, 1]$ is a single hyperparameter to control the addition of an uniform distribution to the unmodified attention weights over the temporal dimension of the input sequence. For both exemplary tasks we investigate the influence of $\gamma$ for relaxing either the encoder self-attention layers (Figures 6a and 7a) or the decoder cross attention layers (Figures 6b and 7b).

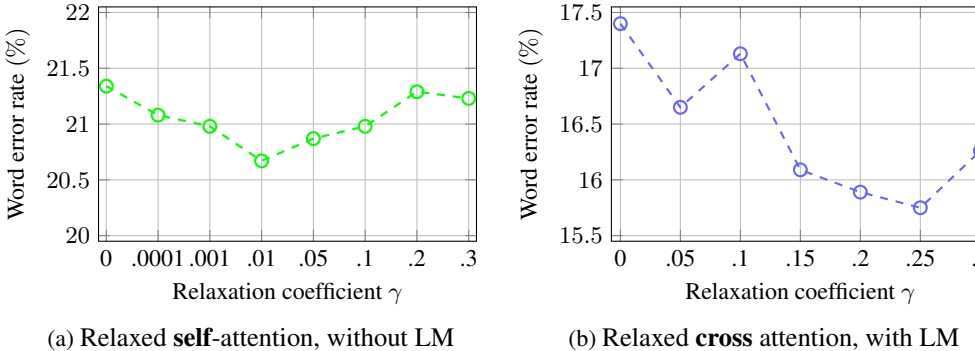

(a) Relaxed **self**-attention, without LM          (b) Relaxed **cross** attention, with LM

Figure 6: Sensitiveness of the **automatic speech recognition** results with respect to the relaxation coefficient $\gamma$ in terms of combined WER (%) on the joint `clean` and `other` portions of the `dev` dataset of the Librispeech task.

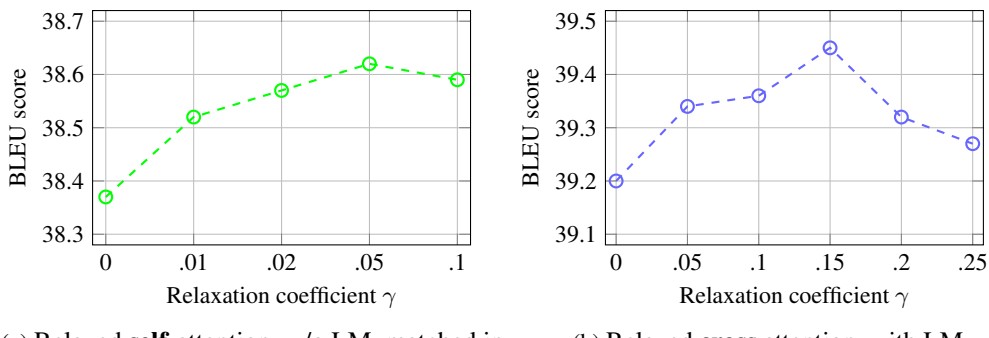

(a) Relaxed **self**-attention, w/o LM, matched in-          (b) Relaxed **cross** attention, with LM
fer.

Figure 7: Sensitiveness of the **neural machine translation** results with respect to the relaxation coefficient $\gamma$ in terms of BLEU scores on the `development` dataset of the IWSLT14 task (DE$\rightarrow$EN).

Both, Figures 6 and 7 show the task-specific performance on the respective development sets, that were used for optimization of the $\gamma$ hyperparameter. In all shown cases, we make the following observations: (i) While relaxed self-attention performs best with smaller $\gamma$ values, relaxed cross attention reaches best performance with somewhat higher values, (ii) there is a smooth and substantial range where relaxed self- and cross attention improves over the resimulated baselines with $\gamma = 0$ thus showing that the contribution of our method is insensitive with respect to the choice of $\gamma$ in these ranges.

## A.8 STATEMENT ON POTENTIAL NEGATIVE SOCIETAL IMPACTS

Our method itself applies to the general transformer model and is—as we have demonstrated— applicable to a variety of applications. Out of these applications, we identify that automatic lip-reading can be used for malicious purposes such as eavesdropping on private civilian conversation in video surveillance footage. The dataset we use for automatic lip-reading consists of professionally recorded speakers that are aware of being recorded, are at a close distance, have well illuminated faces

while speaking, and are mostly facing towards the camera. These conditions are very unlikely in a malicious surveillance scenario where it is unlikely that the methods and models we developed in our work are of large benefit. In addition, we believe that the positive impact of lip-reading applications clearly outweighs the possible negative applications. Examples of such applications are (i) improving speech recognition in case audio is corrupted, (ii) helping in crime investigations, (iii) enabling people suffering from aphonia to communicate, (iv) silence dictations, and (v) uttering silent alarms or passphrases for security.

