# OpenReview forum: "Relaxed Attention for Transformer Models"
_ICLR.cc/2023/Conference — Submitted to ICLR 2023_

### Official Review · Reviewer_4P4f · 2022-10-24

**Confidence:** 4
**Correctness:** 3
**Technical Novelty And Significance:** 2
**Empirical Novelty And Significance:** 2
**Recommendation:** 5

**Clarity, Quality, Novelty And Reproducibility:**

The formulation of the problem is clear and the paper is well-written. However, the contribution is quite limited.

**Strength And Weaknesses:**

**Strengths:**

1. The paper studies various tasks including on text, speech and image, which guarantees that the conclusions are general and robust.
2. The paper is generally well-written and easy to follow.


**Weaknesses:**
1. The contribution is somewhat incremental. Extending relaxed attention from cross-attention to self-attention is straightforward, the paper shows that it will bring some improvements to the model performance, but lacks showing the novelty of doing so. For example,  what it brings to the training/inference of the model? Does it solve some problems that the original attention has? And how the relaxed attention achieves that? In my experience, the sparsity in self-attention is natural and provides clear local context information to the model, then what the smoothing regularization term brings to the model?
2. Another contribution, the matched inference, does not show consistent improvement across different tasks. The paper does not provide a detailed analysis of the results which also limits the novelty. If it cannot bring improvements and no in-depth explanation is conducted, then why take it as one of the contributions?
3. The method section is too short to provide the details and insights of the proposed method.
4. The experimental settings lack some details. For example, what is the difference between the clean and other settings in the ASR task? In the “Approach” column in each table, is the relaxed self-attention added on top of relaxed cross-attention or they are separately enabled? In the machine translation tasks, have you conducted experiments on larger benchmark datasets such as WMT which will make the conclusions more faithful?


**Summary Of The Paper:**

The paper studies the relaxed attention method on the Transformer architecture, across a variety of tasks including automatic speech recognition, lip reading, machine translation, image classification. The technical contributions include exploring relaxed attention on the self-attention module, utilizing it on both training and inference, and introducing variation into image classification where the sequence length is fixed.

**Summary Of The Review:**

I believe the novelty of the paper is straightforward, but the contribution is limited.

---

> ### Author Response · Authors · 2022-11-09
> **Response to Reviewer 4P4f**
>
> **Regarding “weakness #1”:**
>
> We understand relaxed self-attention as a regularization method employing a manipulation of the attention weights during training to “introduce some stress into the training process” as we mention in the paper in Sec. 3, Par. 2. In other words, we deliberately make the softmax outputs noisy during training to provide regularization to the model. In addition, please note that the normalizing division by $T$ in eq. (1) is different for each file and minibatch (for the sequence tasks). Accordingly, the varying $T$ cannot simply be compensated by subsequent layers in the training.
> Without making any assumption of the smoothness of the attention weights during inference, we empirically show the benefit of this method in our results. We will include this line of arguments as further explanation of the method into the final paper and will make a deeper analysis of attention-weight entropy during inference part of our future research.
>
> **Regarding “weakness #2”:**
>
> The manipulation introduced by relaxed attention creates some mismatch between training and test, which might also be harmful for the model. This motivated us to see whether avoiding such mismatch affects performance, some methods e.g., in ASR also use smoothing during inference as it can help during decoding (e.g., in the reference method “smoothed focus” from Chorowski et al. (2015) smoothing is done during training *and* inference and we report this baseline in all main result tables 1-4), but as you correctly observed, it only helps in a few cases. In a nutshell: The matched inference case is more meant and presented as a consistent ablation study than as a contribution. However, if the reviewer feels more comfortable with moving the matched inference case to the Appendix, we offer to do this.
>
> **Regarding “weakness #3”:**
>
> Due to the page limitation of the main paper, some of the requested in-depth analyses are in the Appendix of the paper. There you find a sensitivity analysis of the relaxation coefficient $\gamma$ (Figure 7), a study on the interaction between relaxed attention and attention dropout (Table 8), as well as a comprehensive investigation on external language model integration with relaxed cross attention (Tables 5 & 6). Among the most important findings, the analysis in the Appendix A.4 reveals that relaxed cross attention weakens the bias from the internally learned LM towards the known training transcriptions, which then opens the door for a profitable combination with a stronger external LM that has been trained with additional (text) data. We hope that all this adds to the methods section and the requested provision of more details.
>
> **Regarding “weakness #4”:**
>
> Thanks for your remarks, which are easily addressable and solvable for us. For example, being more specific about the tasks is really important, as we cover a variety of tasks which might be unknown to the readers. We will carefully review the experimental settings section and add missing information where it is required. For the specific case of the ASR task, `clean` denotes a set of speech files that are mostly free from background noise and recorded under controlled conditions with quality microphones. The `other` set contains more challenging recordings that are noisy in various aspects (background noise, reverberation, artifacts, clipping, etc.). In the “Approach” column in our result tables, relaxed self- and cross attention are always separately enabled. We also thank the reviewer for the helpful suggestion on using WMT and take this as a strong recommendation to follow up in our future research work.
>
> **Novelty and contribution:**
>
> Please note that reviewer #1 correctly mentioned that the “work shows better or on par performance with state-of-the-art models on lip-reading LRS3 benchmark (-0.6% in word error rate) and machine translation IWSLT14 (DE->EN) (+0.18 BLEU)”. We would kindly remind the reviewer that the approach from Lohrenz et al., (2021), only proposed relaxed cross attention for the ASR task. In our work, however, we introduced relaxed self-attention, which might use the same mathematical manipulation, but is still an unusual step to go, and no straightforward extension. This is, because Lohrenz et al. (2021) were not able to achieve improvements in the decoder *without* an additional language model. On the two tasks LRS3 and IWSLT14, which reviewer #1 mentioned, we provide new top-performing methods using our introduced relaxed self-attention, but without using a language model! Isn’t this surprising and accordingly a certain step regarding novelty?

---

### Official Review · Reviewer_2CCe · 2022-10-25

**Confidence:** 2
**Correctness:** 2
**Technical Novelty And Significance:** 2
**Empirical Novelty And Significance:** 2
**Recommendation:** 5

**Clarity, Quality, Novelty And Reproducibility:**

The paper is overall clear.

It introduces a simple idea with a lot of experimental results. The significance of the proposed model is concerning as the improvement is marginal.

It is somewhat novel, although similar idea was proposed in literature, extending the application to wider range is still valuable.

The code is provided in supplementary.

**Strength And Weaknesses:**

Strength:
1. Simple techniques apply to wide range of application.
2. Extensive experiments.

Weakness
1. Novelty. The method was first proposed in [1] as the authors cited in paper.
2. The experimental results are not strong enough as the improvement seems marginal. And also lack of variance on all results. For the results with less significant improvement, it would better to also show the variance among runs.

[1] TimoLohrenz,PatrickSchwarz,ZhengyangLi,andTimFingscheidt.RelaxedAttention:ASimple Method to Boost Performance of End-to-End Automatic Speech Recognition. InProc. of ASRU, pp. 177–184, Cartagena, Colombia, December 2021.

**Summary Of The Paper:**

In this paper, the authors proposed to use relaxed attention in both self attention and cross attention part of transformer models. In details, the authors proposed to add a smoothing term on the cross attention module with a fuzzy realaxation coefficient drawn from a Gaussian distribution. Extensive experiments were done on automatic speech recognition, lip-reading, machine translation, and image classification.

**Summary Of The Review:**

Generally I think relaxed attention is an interesting idea, as it is simple and easy to apply in multiple applications. However, the efficiency of the technique is supported by experimental results, which is not significant enough as the improvements are marginal and metrics' variance between runs are not provided.

---

> ### Author Response · Authors · 2022-11-09
> **Response to Reviewer 2CCe**
>
> Thanks for your valuable comments and acknowledging the extensiveness of our experiments across a variety of tasks and applications.
>
> **Novelty:**
>
> Please note that reviewer #1 correctly mentioned that the “work shows better or on par performance with state-of-the-art models on lip-reading LRS3 benchmark (-0.6% in word error rate) and machine translation IWSLT14 (DE->EN) (+0.18 BLEU)”. We would kindly remind the reviewer that the “method […] first proposed in” (Lohrenz et al., 2021) only proposed relaxed cross attention for the ASR task. In our work, however, we introduced relaxed self-attention, which might use the same mathematical manipulation, but is still an unusual step to go. This is, because Lohrenz et al. (2021) were not able to achieve improvements in the decoder *without* an additional language model. On the two tasks LRS3 and IWSLT14, which reviewer #1 mentioned, we provide new top-performing methods using our introduced relaxed self-attention, but without using a language model! Isn’t this surprising and accordingly a certain step regarding novelty?
>
> **Regarding experimental results:**
>
> We fully agree that a significance analysis of the results is important. As reading the Appendix is not mandatory for reviewers, we want to point the reviewer to our Appendix A.5, Table 7, where we provided a sensitivity analysis using five different initializations for all four investigated applications and report average metrics with their standard deviations. Due to limited computational resources, we selected the best methods among the novel relaxed self-attention approaches and compared to the respective baseline. Statistical significance is shown for ASR, for lip-reading, and for machine translation. For image classification, improvements have been shown but just not reaching significance level. We are happy to leverage that analysis into the main paper to convince the reader about the usefulness of our method.

---

### Official Review · Reviewer_WiHX · 2022-10-25

**Confidence:** 4
**Correctness:** 3
**Technical Novelty And Significance:** 2
**Empirical Novelty And Significance:** 2
**Recommendation:** 3

**Clarity, Quality, Novelty And Reproducibility:**

- The paper is clearly written.
- It falls short in terms of technical novelty.
- Without carefully checking the appendix, my educated guess is that the results are fairly easy to reproduce.

**Strength And Weaknesses:**

Strength
- Clear presentation
- The method is straightforward
- Experiments cover a variety of tasks

Weaknesses
- Extending an existing technique to self and causal attention is thin in terms of technical contribution
- No evidence is provided to support the significance of the improvements (if any)
- Lacks in-depth analysis and discussion of the method
- An important baseline is missing: smoothing the attention weights using a larger temperature
- The “relaxed attention” naming is unfortunate—it leaves me the impression that it is lifting some constraints in conventional attention (e.g., simplex). I suggest changing it to something related to “smoothed” or “flat.”

Details
- The improvements in the MT experiments seem very marginal—the ~0.2 BLEU delta is definitely reachable with a lucky random seed; I suspect this might be the case in other tasks too (MT is the only one I’m familiar with). The paper can benefit from training multiple models with different random seeds, and/or presenting significance tests.
- It is great to cover a diverse set of tasks, but the paper favors width over depth. I would really appreciate it if the paper could include some in-depth analysis and discussion of the method. For example, how does the smoothing term affect the “flatness” of attention? Comparing the entropy of the attention weights w/ and w/o the uniform term is a good starting point. Performance change over different choices of gamma is another good one. Some discussion/analysis on why smoother attention is better is also interesting.

**Summary Of The Paper:**

This paper explores relaxed attention in transformer models. The relaxed attention interpolates the attention weights with a uniform distribution. This method was proposed by previous work, and this paper extends its application from cross attention to all attention use cases (cross, self, causal). The paper further proposes a variant that randomly draws the interpolating coefficients at training time.

The experiments cover a variety of tasks, including automatic speech recognition, lip reading, machine translation, and image classification. On some, the results are mixed; on others, the improvements are marginal.



**Summary Of The Review:**

The paper did a good job in terms of covering a variety of tasks. But it lacks depth at the same time. Given that the technical contribution is thin compared to average ICLR papers, I suggest that the authors should dive deeper into the core assumption of the paper: smooth attention is better in some applications.

---

> ### Author Response · Authors · 2022-11-09
> **Response to Reviewer WiHX**
>
> Thanks for your valuable comments. Below, we address your main concerns.
>
> **Regarding evidence to support the significance of the improvements:**
>
> We fully agree that a significance analysis of the results is important. As reading the Appendix is not mandatory for reviewers, we want to point the reviewer to our Appendix A.5, Table 7, where we provided a sensitivity analysis using five different initialization seeds for all four investigated applications and report average metrics with their standard deviations. Due to limited computational resources, we selected the best methods among the novel relaxed self-attention approaches and compared to the respective baseline. Statistical significance is shown for ASR, for lip-reading, and for machine translation. For image classification, improvements have been shown but just not reaching significance level. We are happy to leverage that analysis into the main paper to convince the reader about the usefulness of our method.
>
> **Regarding in-depth analysis and discussion, different choices of $\gamma$:**
>
> As you positively mentioned, the scope of the main body is intended to show the versatility of the method on a variety of tasks. Due to the page limitation of the main paper, some of the requested in-depth analyses are in the Appendix of the paper, among these also your requested sensitivity analysis of the relaxation coefficient $\gamma$ in Appendix A.7, Figures 6 & 7. We hope that you acknowledge this. In addition, we also empirically investigated the interaction between relaxed attention and attention dropout (Table 8), as well as a comprehensive investigation on external language model integration with relaxed cross attention (Tables 5 & 6). Among the most important findings, the analysis in the Appendix A.4 reveals that relaxed cross attention weakens the bias from the internally learned LM towards the known training transcriptions, which then opens the door for a profitable combination with a stronger external LM that has been trained with additional (text) data.
>
> **Regarding the missing baseline using tempered softmax for attention:**
>
> Indeed, tempering the softmax function has a smoothing effect. In early experiments on a different ASR task, we also performed experiments with temperature coefficients >1 and could not observe any benefit from this type of smoothing. As we are not aware of any literature, where tempered softmax in the attention weight matrix helps for single-model transformer training, we instead chose the smooth focus method from Chorowski et al. (2015), where the attention weight distribution is similarly smoothed by using a sigmoid function instead of an unbound exponential function in the softmax. Please understand that we consider this as sufficient baseline.
>
> **Technical novelty:**
>
> Please note that reviewer #1 correctly mentioned that the “work shows better or on par performance with state-of-the-art models on lip-reading LRS3 benchmark (-0.6% in word error rate) and machine translation IWSLT14 (DE->EN) (+0.18 BLEU)”. We would kindly remind the reviewer that the “existing technique” (Lohrenz et al., 2021) only proposed relaxed cross attention for the ASR task. In our work, however, we introduced relaxed self-attention, which might use the same mathematical manipulation, but is still an unusual step to go. This is, because Lohrenz et al. (2021) were not able to achieve improvements in the decoder *without* an additional language model. On the two tasks LRS3 and IWSLT14, which reviewer #1 mentioned, we provide new top-performing methods using our introduced relaxed self-attention, but without using a language model! Isn’t this surprising and accordingly a certain step regarding novelty?

---

> > ### Comment · Reviewer_WiHX · 2022-11-18
> > **After author response**
> >
> > Thanks for pointing me to the results in the Appendices. The response addressed my concerns about the analysis on $\gamma$. However, my other concerns remain. I keep my score unchanged.

---

### Official Review · Reviewer_ztHw · 2022-10-25

**Confidence:** 3
**Correctness:** 4
**Technical Novelty And Significance:** 2
**Empirical Novelty And Significance:** 3
**Recommendation:** 5

**Clarity, Quality, Novelty And Reproducibility:**

The paper is clear, with minor places where the fluency of the sentence could be improved.
The paper conducts experiments that introduces new understanding into a previously proposed method.
I believe the experiments can be reproduced.

**Strength And Weaknesses:**

Strength:
1. The experiments are detailed and cover multiple tasks/domains (speech, translation, and image).

Weakness:
Minor sentence fluency:
1. page 2: "Very early once ..."
2. page 9 in Conclusions: "thereby regularizing already in the encoder"

Question:
  - Have you tried applying the relaxed attention in the pretraining stage?

**Summary Of The Paper:**

The paper conducts detailed study on relaxed attention for transformer. The relaxed attention mechanism is applying smoothing to the attention weights. Previous work which proposed relaxed attention only applied it to cross-attention in the decoder. This work studies applying relaxed attention to self-attention layer in the encoder. The authors suggest two benefits: 1. the relaxed attention provides regularization when applied to the self-attention layer. 2. the relaxed attention applied to the cross-attention layer suppresses the intrinsic language model and allows better integration of an external language model. The experiments are conducted in automatic speech recognition (ASR), lip-reading, machine translation, and image classification. The work shows better or on par performance with state-of-the-art models on lip-reading LRS3 benchmark (-0.6% in word error rate) and machine translation IWSLT14(DE->EN) (+0.18 BLEU).

**Summary Of The Review:**

The thorough experiments confirm the usefulness of an previously proposed technique that smooths the attention weights.

---

> ### Author Response · Authors · 2022-11-09
> **Response to Reviewer ztHw**
>
> Thank you for your valuable comments and especially the remarks on sentence fluency, we will rephrase the sentences as follows:
>
> 1. “Very early once attention-based encoder-decoder networks were introduced to ASR,…” $\rightarrow$ “When attention-based encoder-decoder networks were first applied to ASR,…”
> 2. "..., thereby regularizing already in the encoder..." $\rightarrow$ “…, thereby providing regularization also in the transformer encoder…”
>
> In addition, we will let the text being proof-read by an English native speaker.
>
> **Regarding your question on the application in the pretraining stage:**
>
> Indeed, including relaxed attention into the pretraining stage is an interesting idea, which we have not investigated so far. Especially the interaction between the random input masking used during pretraining and relaxed attention raises interesting questions. However, pretraining requires large amounts of computational resources (especially when optimizing $\gamma$ values with multiple pretrainings), which were already limited as for the scope of this paper we conducted an extensive evaluation on four quite different tasks for transformer models.
>
> Please note that for the lipreading task we use pretrained models and show that our relaxed attention methods work very well also on large and already pre-trained models.
>
> **On novelty:**
>
> Thank you very much to explicitly mention that “The work shows better or on par performance with state-of-the-art models on lip-reading LRS3 benchmark (-0.6% in word error rate) and machine translation IWSLT14 (DE->EN) (+0.18 BLEU).” In the novelty part, however, you only acknowledge a “new understanding into a previously proposed method”. We would kindly remind the reviewer that such previously proposed method (Lohrenz et al., 2021) only proposed relaxed cross attention for the ASR task. In our work, however, we introduced relaxed self-attention, which might use the same mathematical manipulation, but is still an unusual step to go. This is, because Lohrenz et al. (2021) were not able to achieve improvements in the decoder *without* an additional language model. On the two tasks LRS3 and IWSLT14 which you mentioned we provide new top-performing methods using our introduced relaxed self-attention, but without using a language model! Isn’t this surprising and accordingly a certain step regarding novelty?

---

### Decision · Program_Chairs · 2023-01-20

**Decision:**

Reject

**Justification For Why Not Higher Score:**

- lack of novelty, limited empirical benefit.


**Justification For Why Not Lower Score:**

- the paper is clear, the approach is simple and easy to incorporate.


**Metareview: Summary, Strengths And Weaknesses:**

This paper extends relaxed attention to self and causal attention. Although relaxed attention was introduced earlier, its potential for translation, image classification and lip reading was not assessed before.

Strength:
- detailed experiments in multiple domains.
- clear paper, easy to follow. It provides supporting code.

Weaknesses:
- the reviewers expressed doubts about the significance of the technical contribution.
- the reviewers found the empirical benefit of the method limited.